NMDAR in bladder smooth muscle is not a pharmacotherapy target for overactive bladder in mice

Xie Xiang 1
Luo Chuang 1
Liang Jia Yu 1
Huang Run 1
Yang Jia Li 1
Li Linlong 1
Li YangYang 1
Xing Hongming 1
Chen Huan huanchen@swmu.edu.cn 1 2
1 Public Center of Experimental Technology and The School of Basic Medical Sciences, Southwest Medical University , Luzhou , Sichuan , China
2 Department of Medicine, Beth Israel Deaconess Medical Center and Harvard Medical School , Boston , MA , United States of America
Frauscher Ferdinand
Electronic publication date: 2021 Jul 7
Publication date: 2021
Volume: 9
Electronic Location ID: e11684
Received 2021 Mar 10; Accepted 2021 Jun 7
Copyright: ©2021 Xie et al.
Copyright year: 2021
Copyright holder: Xie et al.
License: This is an open access article distributed under the terms of the Creative Commons Attribution License, which permits unrestricted use, distribution, reproduction and adaptation in any medium and for any purpose provided that it is properly attributed. For attribution, the original author(s), title, publication source (PeerJ) and either DOI or URL of the article must be cited.
License URL: https://creativecommons.org/licenses/by/4.0/

Keywords: Bladder smooth muscle, Cyclophosphamide, Cystometry, MK801, N-methyl-D-aspartate receptor, Overactive bladder, Voiding spot assay

Funding: National Natural Science Foundation of China 81800670 Luzhou City Bureau of Science and Technology 2020LZXNYDZ05 Undergradute Innovation and Enterpreneurship Training Program S201910703072 202010632005 2020360 This work was supported by grants from the National Natural Science Foundation of China (81800670), Luzhou City Bureau of Science and Technology (2020LZXNYDZ05), Undergradute Innovation and Enterpreneurship Training Program (S201910703072, 202010632005, 2020360). The funders had no role in study design, data collection and analysis, decision to publish, or preparation of the manuscript.

==============================
Overactive bladder (OAB) is a common condition that affects a significant patient population. The N-methyl-D-aspartate receptor (NMDAR) has a role in developing bladder overactivity, pharmacological inhibition of which inhibits bladder overactivity. The common pathogenesis of OAB involves bladder smooth muscle (BSM) overactivity. In this study, a smooth muscle–specific NMDAR knockout (SMNRKO) mouse model was generated. The bladders from SMNRKO mice displayed normal size and weight with an intact bladder wall and well-arranged BSM bundles. Besides, SMNRKO mice had normal voiding patterns and urodynamics and BSM contractility, indicating that NMDAR in BSM was not essential for normal physiological bladder morphology and function. Unexpectedly, cyclophosphamide (CYP)-treated SMNRKO and wild-type (WT) mice had similar pathological changes in the bladder. Furthermore, SMNRKO mice displayed similar altered voiding patterns and urodynamic abnormalities and impaired BSM contractility compared with WT mice after CYP treatment. MK801 partially reversed the pathological bladder morphology and improved bladder dysfunction induced by CYP, but did not cause apparent differences between WT mice and SMNRKO mice, suggesting that NMDAR in BSM was not involved in pathological bladder morphology and function. Moreover, the direct instillation of NMDAR agonists or antagonists into the CYP-induced OAB did not affect bladder urodynamic function, indicating that NMDAR in BSM was not the pharmacotherapy target of MK801 for CYP-induced cystitis. The findings indicated that NMDAR in BSM was not essential for normal physiological or pathological bladder morphology and function, and MK801 improving pathological bladder function was not mediated by an action on NMDAR in BSM.

Introduction

The International Continence Society defines overactive bladder (OAB) as a complex of symptoms characterized by the urgency with or without urge incontinence, usually with frequency and nocturia (Abrams et al., 2003). OAB is a common condition in the US, with a reported prevalence of 16% for men and 16.9% for women (Stewart et al., 2003). In Asia, the prevalence of OAB is 19.5% in men and 22.1% in women (Chuang et al., 2019). The symptoms associated with OAB can significantly impair the quality of life and affect the social, psychological, occupational, domestic, physical, and sexual aspects of those who suffer from it. Current pharmacotherapies for OAB include antimuscarinic and β3-adrenoceptor agonist medications. However, these medications have either limited efficacy or significant side effects and hence many patients with OAB reject these drugs (Andersson & Wein, 2004; Chapple et al., 2008). The N-methyl-D-aspartate receptor (NMDAR) is involved in developing bladder overactivity (Ishida et al., 2003; Tanaka et al., 2003; Liu et al., 2015). The pharmacological blockage of NMDAR using the noncompetitive antagonist MK801 inhibits bladder overactivity and improves bladder function (Ishida et al., 2003; Tanaka et al., 2003; Liu et al., 2015). Improvements in the understanding of NMDAR in bladder function may facilitate the discovery of new pharmacotherapies for OAB.

NMDAR is overexpressed and overactivated within irritable bowel syndrome and OAB and mediates visceral hyperactivity (Gaudreau & Plourde, 2004; Yokoyama, 2010). The common pathogenesis of irritable bowel syndrome and OAB involves smooth muscle overactivity (Cukier, Cortina-Borja & Brading, 1997; Kanazawa et al., 2014). The role of intracellular calcium is of particular importance in smooth muscle overactivity, as it is the main determinant of smooth muscle contractile activity. NMDAR has an ionotropic property that regulates calcium influx and calcium-dependent physiological effects. NMDAR in airway smooth muscle (ASM) mediates airway contractile responses and airway hyperreactivity (Anaparti et al., 2015), overactivation of which promotes vascular remolding and pulmonary arterial hypertension (PAH) (Dumas et al., 2018). However, the NDMAR knockout in smooth muscle cells or the pharmacological inhibition of NMDAR has beneficial effects on cardiac and vascular remodeling and improves PAH (Dumas et al., 2018). However, the role of NMDAR in bladder smooth muscle (BSM), thereby regulating bladder function, has not been investigated.

A smooth muscle–specific NMDAR knockout (SMNRKO) mouse model was generated in this study by mating Grin1 fl∕fl (Grin1 encoding GluN1) mice with Sm22 α-cre mice to determine the role and effect of NMDAR in BSM function. As all of the major NMDAR heterotetramers are thought to contain two GluN1, this strategy theoretically created an NMDAR-null in BSM. The bladder of this mouse model was then investigated by the combination of void spot assay (VSA), cystometrogram (CMG), myography, and morphological approaches. The study further examined the role of NMDAR in BSM under pathological conditions. Cyclophosphamide (CYP) was used to induce cystitis in wild-type mice and SMNRKO mice, and comparative studies between these two mouse models were performed. Moreover, the study investigated the effects of NMDAR agonists and antagonists on CYP-induced cystitis through action on NMDAR in the bladder. The study provided evidence that NMDAR in BSM was not essential for normal physiological and pathological bladder morphology and function, and MK801 improving pathological bladder function was not mediated by the inhibition of NMDAR in BSM.

Material and Methods

Animals and reagents

Smooth muscle NMDAR homozygous knockout mice were generated by crossing mice carrying Cre recombinase under the control of the smooth muscle transgelin promoter (SM22 α-creKI, Stock 006878; Jackson Laboratory) with Grin1 fl∕fl mice (Stock 005246; Jackson Laboratory) and then interbreeding the offspring. SM22 α-creKI mice and Grin1 fl∕fl mice were purchased from Jackson Laborarory (Bar Harbor, ME, USA). Wild-type (WT) mice were purchased form SiPeiFu Co., Ltd (Beijing, China). All mice used in this study were in the C57BL/6J background and were aged 10–12 weeks. The mice were housed in standard polycarbonate cages and 5 mice per cages. The mice house environment was maintained on a 12:12-h light–dark cycle at 25 °C with humidity between 40%–70%. The mice in cages had free access to standard laboratory food and water. Criteria for euthanizing animals before the end of the study were as follows. Under normal physiological conditions, any animals showing dying or sickness will be euthanized. Animal receiving intraperitoneal dosing will be monitored, animals showing signs of mis-dosing or dying will be euthanized. Euthanasia was performed by introduction of 100% carbon dioxide into a bedding-free cage initially containing room air with the lid closed at a rate sufficient to induce rapid anesthesia, with death occurring within 2.5 min. All animal care and experimental procedures were performed in adherence to the National Institutes of Health Guidelines for the Care and Use Committee of the Southwest Medical University (20180307002). Unless otherwise specified, all chemicals were obtained from Sigma (St. Louis, MO, USA) and were of reagent grade or better.

Experiment design

To evaluate the voiding function of SMNRKO mice under normal physiological conditions, female WT mice (n = 16), female SMNRKO mice (n = 22), male WT mice (n = 13), and male SMNRKO mice (n = 12) were subjected to voiding spot assay (VSA) on filter paper. Female WT mice (n = 5) and female SMNRKO mice (n = 5) were then assessed for urodynamic function by CMG study. To further determine the BSM contractility of SMNRKO mice, myography was performed on isolated strips from WT mice and SMNRKO mice. Isolated strips from female WT mice (n = 6), isolated strips from female SMNRKO mice (n = 6), isolated strips from male WT mice (n = 8), and isolated strips from male SMNRKO mice (n = 8) were subjected to electrical field stimulation (EFS), KCl stimulation, α, β-meATP stimulation, and carbachol stimulation, respectively. Additional mice were also euthanized for Masson’s trichrome staining (n = 3 for female WT mice, n = 3 for female SMNRKO mice, n = 3 for male WT mice and n = 3 for male SMNRKO mice) and immunofluorescence staining (n = 3 for female WT mice, n = 3 for female SMNRKO mice, n = 3 for male WT mice and n = 3 for male SMNRKO mice).

To study the role and effects of NMDAR in BSM under pathological conditions, cystitis was induced in female WT and SMNRKO mice by intraperitoneal CYP injection at a single dose of 300 mg kg−1 body weight. Immediately after CYP injection, the MK801 treated cystitis group was administered with MK801 at a dose of 3 mg kg−1 body weight. The mice survived for 48 h and were subsequently subjected to void spot assay (VSA), cystometrogram (CMG), myography, and Masson’s trichrome staining studies. Specifically, Female WT mice treated with CYP (WT-CYP, n = 10), female SMNRKO mice treated with CYP (SMNRKO-CYP, n = 10), female WT mice treated with CYP and MK801 (WT-CYP-MK801, n = 12), and female SMNRKO mice treated with CYP and MK801 (SMNRKO-CYP-MK801, n = 11) were subjected VSA on filter paper. WT-CYP mice (n = 5), SMNRKO-CYP mice (n = 5), WT-CYP-MK801 mice (n = 5), and SMNRKO-CYP-MK801 mice (n = 5) were then assessed for urodynamic function by CMG. Isolated strips from WT-CYP mice (n = 9), isolated strips from SMNRKO-CYP mice (n = 9), isolated strips from WT-CYP-MK801 mice (n = 9), and isolated strips from SMNRKO-CYP-MK801 mice (n = 9) were subjected to electrical field stimulation (EFS), KCl stimulation, α, β-meATP stimulation, and carbachol stimulation, respectively. The remaining mice were euthanized for Masson’s trichrome staining (n = 3 for WT-CYP mice, n = 3 for SMNRKO-CYP mice, n = 3 for WT-CYP-MK801 mice, and n = 3 for SMNRKO-CYP-MK801 mice).

To study whether the direct instillation of NMDAR agonists or antagonists into the bladder had effects on bladder urodynamic function, CMG studies were performed on female WT mice pretreated with CYP during the bladder infused with NMDAR agonists or antagonists, including 100 µM NMDA (n = 4), 100 µM RS-tetrazol-5-yl glycine (n = 4), 100 µM D-AP5 (n = 4), 100 µM CGS 19755 (n = 4), and 100 µM MK 801 (n = 6).

Void spot assay (VSA)

Individual mice were gently placed in a standard mouse cage with Blicks Cosmos Blotting Paper (Catalog no. 10422-1005) placed at the bottom for 4 h, during which time water was withheld and standard dry mouse chow was available. The mice were then returned to their home cages, and the filter paper was recovered. The same individual mice were then used to repeat the experiment the next day. Filters were imaged under ultraviolet light at 365 nm in a UVP Chromato-Vue C-75 system (UVP, CA, USA) with an onboard Canon digital single-lens reflex camera (EOS Rebel T3, 12 megapixels). Overlapping voiding spots were visually examined and manually separated by outlining and copying, and then pasted on to a nearby empty space using ImageJ software (http://fiji.sc/wiki/index.php/Fiji). The images were analyzed using UrineQuant software developed in collaboration with the Harvard Imaging and Data Core. The result table containing the area of each individual voiding spot and the total number of spots was imported into Excel for statistical analysis. A volume:area standard curve showed that a 1-mm2 voiding spot represents 0.283 µL of urine. Voiding spots with an area ≥ 80 mm2 were considered primary voiding spots (PVS) based on voiding spot patterns from hundreds of mice (Chen et al., 2017; Rajandram et al., 2016).

Cystometrogram (CMG)

CMG was performed with PBS infusion (25 µL min−1) as previously described (Hao et al., 2019). The mice were anesthetized by subcutaneously injecting urethane (1.4 g kg−1 from 250 mg mL−1 solution in PBS) 30–60 min before the surgery. At the time of surgery, the mice were further anesthetized with continuous-flow isoflurane (3% induction, 1.0% maintenance). Once the pedal reflex was absent, a 1-cm midline abdominal incision was performed. Flame-flanged polyethylene-50 tubing sheathing a 25G ×1.5 in. needle was implanted through the dome of the bladder and then sutured in place (8-0 silk purse-string sutures). The incision site was sutured around the tubing using sterile 5-0 silk sutures, and the mice were then placed into a Bollman mouse restrainer for 30- to 60-min stabilization. The catheter was connected to a pressure transducer (and syringe pump by the side arm) coupled to data-acquisition devices (WPI Transbridge and AD Instruments PowerLab 4/35) and a computerized recording system (AD Instruments LabChart software). Bladder filling then commenced, following which voiding occurred naturally through the urethra. Repeated voiding cycles were assessed for the change in the voiding interval (time between peak pressures), basal pressure (minimum pressure after voiding), threshold pressure (pressure immediately before the onset of voiding contraction), peak pressure (maximum voiding pressure minus basal pressure), and compliance (volume, in µL required to increase pressure by one cm H2O). 

Myograph

Bladders were pinned on a small Sylgard block, and the muscle was dissected free of the mucosal tissue. BSM strips were then cut longitudinally (two mm wide and seven mm long). Muscle strips were mounted in an SI-MB4 tissue bath system (World Precision Instruments, FL, USA). Force sensors were connected to a TBM 4M transbridge (World Precision Instruments), and the signal was amplified by PowerLab (AD Instruments, CO, USA) and monitored through Chart software (AD Instruments). BSM strips were gently prestretched to optimize contraction force and then pre-equilibrated for at least 1 h. All experiments were conducted at 37 °C in physiological saline solution (in mM: 120 NaCl, 5.9 KCl, 1.2 MgCl2, 15.5 NaHCO3, 1.2 NaH2PO4, 11.5 glucose, and 2.5 mM CaCl2) with continuous bubbling of 95% O2/5% CO2. Contraction force was sampled at 2000/s using Chart software. BSM tissue was treated with agonists or antagonists and/or subjected to electrical field stimulation (EFS). BSM strip EFS was carried out using a Grass S48 field stimulator (Grass Technologies, RI, USA) using previously described standard protocols (Rajandram et al., 2016; Chen et al., 2020).

Histological and immunofluorescence staining

For histopathological evaluation, the bladders were removed under anesthesia, fixed in 4% formaldehyde, embedded in paraffin blocks, and sectioned. The sections were stained with Masson’s trichrome staining. For immunofluorescence staining, excised bladders were fixed in 4% (w/v) paraformaldehyde for 2 h at room temperature. The fixed tissue was cryoprotected, frozen, sectioned, and incubated with a purified rat anti-mouse β1 integrin antibody (1:100, #550531, BD Bioscience) overnight at 4 °C. The sections were then incubated with an Alexa Fluor 488–conjugated secondary antibody (diluted 1:100), and nuclei were stained with DAPI. Imaging was performed on an Olympus BX60 fluorescence microscope with a 40 ×/0.75 objective. Images (512 and 512 pixels) were saved as TIFF files and imported into an Adobe Illustrator CS3. In the present study, each bladder was sectioned to obtain two slides with approximately four sections of tissue on each slide. Each tissue section was examined under a microscope to ensure the consistency of the staining result, and images were taken and quantitated.

Statistical analyses

All data were presented as boxes and whiskers. The centerline was the median of the data set, the box represented 75% of the data, and the bars indicated whiskers from minimum to maximum. Data were analyzed by the Student t-test between the two groups using GraphPad Prism 8 software, and a P value <0.05 was considered significant.

Results

SMNRKO mice had normal bladder morphology

SM22 α-Cre mice were crossed with Grin1 fl∕fl mice to obtain smooth muscle–specific knockout protein. The deletion of the Grin1 gene was verified by genomic DNA genotyping and mRNA expression analysis (data shown in Supplemental Information). Bladders from female and male SMNRKO mice both displayed normal size and weight (Figs. 1A, 1C). Since SMNRKO mice of both sexes had no obvious changes in body weight (Fig. 1B), the ratios of bladder weight (mg) to body weight (g) revealed no difference between SMNRKO mice and wild-type mice (Fig. 1D). Masson’s trichrome–stained bladder sections from female and male SMNRKO mice were examined to further investigate whether the histology changed in the bladder of SMNRKO mice. As shown in Figs. 1E–1N, the histology of the bladder from both sexes revealed no observed differences between SMNRKO mice and wild-type (WT) mice. The bladder wall of WT mice and SMNRKO mice had similar thickness and integrity; no evidence indicated derangement or separation (Figs. 1E, 1F, 1K, and L). The smooth muscle layer showed compact well-arranged smooth muscle bundles with collagen fibers in between (Figs. 1G, 1H, 1M, and N). The thickness of the smooth muscle layer from both sexes of SMNRKO mice revealed no differences compared with that of WT mice. Furthermore, β-integrin immune-stained sections revealed intact, well-distributed smooth muscle cells with normal size (Figs. 1I, 1J, 1O, and P). In summary, these data suggested that NMDAR in BSM did not contribute to bladder morphology in mice.

Figure 1 Bladders of SMNRKO mice exhibited normal morphology.

(A) Images of female and male bladders from WT and SMNRKO mice. (B, C and D) SMNRKO mice (female, n = 9; male, n = 6) had similar body weight, bladder weight, and bladder-to-body weight ratio compared with WT mice (female, n = 7; male, n = 7). (E, F, G and H) Representative Masson’s trichrome staining images of bladder sections from female WT mice and SMNRKO mice. (I and J) Bladder sections from female WT and SMNRKO mice were subjected to immunofluorescence staining with anti- β1 integrin antibody (green) and DAPI for nuclei (blue). (K, L, M and N) Representative Masson’s trichrome staining images of bladder sections from male WT mice and SMNRKO mice. (O and P) Bladder sections from male WT and SMNRKO mice were subjected to immunofluorescence staining with anti- β1 integrin antibody and DAPI for nuclei, respectively. Data are shown as boxes and whiskers, the centerline is the median of the data set, the box represents 75% of the data, and bars indicate whiskers from minimum to maximum. Data were analyzed using the Student t test. ∗P < 0.05, ∗∗P < 0.001 compared with female or male WT mice.

SMNRKO mice exhibited normal urinary function and BSM contractility

VSA was performed in female and male mice to evaluate whether NMDAR in BSM was involved in regulating bladder voiding function. Female SMNRKO mice produced approximately 3.68 PVS with an average size of ∼357.20mm2/spot in a 4-h period, which showed no significant difference compared with female WT mice (Figs. 2A, 2B, 2E, and F). Besides, they had similar total voiding volume (Fig. 2G). To show shifts in the overall pattern of voiding behavior, all the spots over 40 mm2 were grouped into different sizes with their frequency distribution. Female SMNRKO mice and female WT mice showed a similar voiding pattern (Fig. 2H). The size −frequency chart of voiding spots showed a distinct voiding size distribution between male SMNRKO mice and WT mice (Fig. 2I). However, the VSA of male SMNRKO mice indicated the grossly same PVS number, PVS size, and total voiding volume as that of male WT mice (Figs. 2C–2G), suggesting that male SMNRKO mice had normal bladder voiding function. Next, CMGs were performed on female SMNRKO mice to evaluate bladder urodynamic function. Fig. 2J and 2K showed CMG traces in which time-dependent bladder pressure changes were recorded during continuous filling and emptying cycles in WT mice and SMNRKO mice, respectively. However, none of the analyzed cystometric parameters of SMNRKO mice showed any significant differences compared with those of WT mice. These included voiding interval, basal pressure, micturition threshold pressure, peak pressure, and bladder compliance (Figs. 2L–2P), thus further confirming that SMNRKO mice had normal bladder function. Myography was performed on isolated strips from WT mice and SMNRKO mice to determine whether NMDAR in BSM affected BSM contractility. As shown in Figs. 3A–3F, the contractile force of BSM strips from female and male SMNRKO mice increased with EFS frequency. However, the average force under each EFS frequency showed no significant differences compared with WT mice. Furthermore, BSM strips from SMNRKO mice of both sexes had normal contractility in response to carbachol, a,b-meATP, and KCl depolarization (Figs. 3G–3I). In summary, these data suggested that NMDAR in BSM did not contribute to normal physiological bladder function in mice.

Figure 2 SMNRKO mice exhibited a normal voiding pattern and urodynamic function.

(A, B, C and D) Representative filters showed ultraviolet light–illuminated urine spots from female and male WT mice (female, n = 32 filters; male, n = 26 filters) and SMNRKO mice (female, n = 44 filters; male, n = 24 filters). (E, F and G) Summarized data of the numbers of primary voiding spots (PVS: voiding spot area ≥ 80 mm2), area of PVS, and the total area of voiding spots per filter indicate normal voiding patterns of SMNRKO mice. (H and I) Summarized frequency distribution charts of spot size from female and male SMNRKO mice indicate similar voiding patterns of SMNRKO mice with WT mice. (J and K) Representative CMG traces from female WT and SMNRKO mice. (L, M, N, O and P) Summarized data of CMGs from female WT (n = 5) and SMNRKO (n = 5) mice indicate a normal urodynamic function of SMNRKO mice. Data are shown as box and whiskers, the centerline is the median of the data set, the box represents 75% of the data, and bars indicate whiskers from minimum to maximum. Data were analyzed by the Student t test. ∗P < 0.05, ∗∗P < 0.001 compared with female or male WT mice.

Figure 3 SMNRKO mice had normal BSM contractility.

(A, B, C and D) Representative traces of BSM contraction from female and male WT mice (female, n = 6; male, n = 8) and SMNRKO (female, n = 6; male, n = 7) mice in response to EFS (1, 2, 5, 10, 20, and 50 Hz). (E and F) Summarized data of EFS-stimulated contraction from SMNRKO mice showed similar contraction force as in WT mice. (G, H and I) Summarized data of KCl-induced contraction, α, β-meATP-induced contraction, and carbachol-induced contraction from female and male SMNRKO mice (female, n = 8; male, n = 8) exhibited normal BSM contractility with WT mice (female, n = 6; male, n = 8). Data are shown as box and whiskers, the centerline is the median of the data set, the box represents 75% of the data, and bars indicate whiskers from minimum to maximum. Data were analyzed by the Student t test. ∗P < 0.05, ∗∗P < 0.001 compared with female or male WT mice.

Figure 4 Deficiency of NMDAR in BSM did not impact pathological bladder morphology.

(A) The images of bladders from female WT and SMNRKO mice pretreated with CYP or CYP and MK801. (B, C and D) SMNRKO mice pretreated with CYP (n = 6) or CYP and MK801 (n = 6) had a similar abnormal body weight, bladder weight, and bladder-to-body weight ratio as WT mice pretreated with CYP (n = 6) or CYP and MK801 (n = 6). MK801 reversed the increased bladder weight and the ratio value, but not the decreased body weight induced by CYP in both WT and SMNRKO mice. (E, F, G and H) Representative Masson’s trichrome staining images of bladder cross-sections from a WT mouse pretreated with CYP. (I, J, K and L) Representative Masson’s trichrome staining images of bladder cross-sections from an SMNRKO mouse pretreated with CYP. (M, N, O and P) Representative Masson’s trichrome staining images of bladder cross-sections from a WT mouse pretreated with CYP and MK801. (Q, R, S and T) Representative Masson’s trichrome staining images of bladder cross-sections from an SMNRKO mouse pretreated with CYP and MK801. The urothelium (UR), lamina propria (LP), and bladder smooth muscle (BSM) layers are indicated. The dashed line indicates the layers of UR, LP, or BSM. The black triangles indicate the thin and disrupted urothelium. The black star indicates edema with enlarged spaces in the LP. The black arrowheads in the muscle layer indicate enlarged spaces within muscle bundles. Data are shown as box and whiskers, the centerline is the median of the data set, the box represents 75% of the data, and bars indicate whiskers from minimum to maximum. Data were analyzed by the Student t test. ∗P < 0.05, ∗∗P < 0.001 compared with normal female WT mice and the control WT mice. n.s: No significance (>0.05).

NMDAR in BSM was not essential for pathological bladder morphology

Pharmacological NMDAR blockage using the noncompetitive antagonist MK801 has beneficial effects on cardiac, vascular, and bladder remodeling under pathological conditions (Liu et al., 2015; Dumas et al., 2018). However, whether NMDAR deficiency in BSM had a beneficial role in bladder remodeling under pathological conditions is unclear. To answer this, the present study investigated the morphology of CYP-induced cystitis bladder and MK801-treated cystitis bladder from female WT mice and SMNRKO mice. As shown in Figs. 4A–4D, the bladders from SMNRKO mice and WT mice were significantly larger and heavier after CYP treatment, indicating bladder hypertrophy. Besides, CYP treatment resulted in a markedly reduced body weight and a significant increase in the ratio of bladder weight to body weight (Fig. 4D). However, these changes revealed no differences between SMNRKO mice and WT mice (Figs. 4B–4D). In CYP-treated WT mice and SMNRKO mice, the administration of MK801 reversed bladder weight and size (Figs. 4A–4D), but still no differences were observed between WT mice and SMNRKO mice (Figs. 4A–4D). Masson’s trichrome staining was then used to investigate the bladder pathological changes. The images of stained tissue from SMNRKO mice and WT mice both displayed dramatic structural changes after CYP treatment. As shown in Figs. 4E–4L, the urothelial layer appeared much thinner with the disrupted superficial inner layer (Fig. 4F and 4J). The lamina propria layer was hugely thickened, and the vasculature within the lamina propria was dramatically dilated (Fig. 4G and 4K). The muscle bundles were disorganized with obvious injury, and interstitial edema among muscle bundles was obvious with enlarged spaces (Fig. 4H and 4L). Importantly, the administration of MK801 partially reversed these CYP-induced changes in both WT mice and SMNRKO mice. As shown in Figs. 4M–4T, the urothelial layer appeared much thicker with no obvious disruption (Fig. 4N and 4R). The lamina propria layer was thinner, and the vasculature within the lamina propria was dilated but with a few numbers (Fig. 4O and 4S). The muscle bundles were well organized, and the interstitial edema among muscle bundles was not apparent (Fig. 4P and 4T). Although the administration of MK801 reversed CYP-induced pathological changes in both WT mice and SMNRKO mice, bladders from both mice displayed similar histological changes without any significant differences. In summary, these data suggested that NMDAR in BSM was not essential for bladder morphology under pathological conditions.

NMDAR in BSM was not involved in pathological bladder function and contractility

NMDAR inhibition with MK801 is effective in improving cardiac, vascular, and bladder function under pathological conditions (Liu et al., 2015; Dumas et al., 2018). Therefore, the present study investigated the voiding function of CYP-induced cystitis bladder and MK801-treated cystitis bladder from female SMNRKO mice and WT mice to examine whether NMDAR deficiency in BSM improved pathological bladder function. As shown in Figs. 5A–5G, SMNRKO mice and WT mice both had compromised bladder function after CYP treatment, characterized by significantly smaller spots and increased numbers of individual void spots. However, no marked differences were observed in analyzed VSA parameters between WT mice and SMNRKO mice after CYP treatment (Figs. 5A–5G). Besides, the size −frequency chart of voiding spots from both mice showed the same altered voiding pattern, which reached the top in an area of 40–80 mm2 and then decreased with the increase in the area (Fig. 5H). In CYP-treated WT mice and SMNRKO mice, all these voiding abnormalities were abrogated after MK801 treatment, as manifested by decreased numbers of individual void spots and increased void spot size (Figs. 5C–5G). In addition, voiding spots of both mice after MK801 treatment showed a relatively normal voiding pattern similar to those in normal WT mice and SMNRKO mice (Fig. 5H). However, the MK801-treated SMNRKO mice still displayed no difference in voiding function compared with MK801-treated WT mice (Figs. 5C–5H), indicating that NMDAR in BSM did not contribute to bladder function under pathological conditions, which was further supported by CMGs. As shown in Figs. 5I–5Q, CYP-treated WT mice and SMNRKO mice both exhibited similar urodynamic abnormalities with decreased voiding interval, peak pressure, and compliance and increased basal pressure (Figs. 5M–5Q). After MK801 treatment, both mice retained a relatively normal urodynamic function with similar normal cystometric parameters (Figs. 5M–5Q). Further, myography was performed to determine whether NMDAR deficiency in BSM affected BSM contractility under pathological conditions. Figs. 6A–6H shows that BSM strips from SMNRKO mice and WT mice after CYP treatment had impaired contractility in response to EFS, carbachol, and KCl depolarization. However, the impaired contractility was reversed by MK801 treatment. Consistently, no significant differences were found in contractility response between SMNRKO mice and WT mice after MK801 treatment (Figs. 6A–6H). In summary, these data suggested that NMDAR in BSM did not contribute to pathological bladder function and contractility.

Figure 5 Deficiency of NMDAR in BSM affected pathological bladder voiding pattern and urodynamic function.

(A, B, C and D) Representative filters showing UV light–illuminated urine spots from female WT mice (CYP, n = 20 filters; CYP-MK801, n = 24 filters) and SMNRKO (CYP, n = 20 filters; CYP-MK801, n = 22) mice pretreated with CYP or CYP and MK801. (E, F, G and H) Summarized data of numbers of PVS, area of PVS, total area of void spots per filter, and frequency distribution of spot size indicate similar altered voiding patterns between WT and SMNRKO mice pretreated with CYP or CYP and MK801. MK801 corrected the increased numbers of PVS, decreased the area of PVS, and altered voiding patterns induced by CYP in both WT and SMNRKO mice. (I, J, K and L) Representative CMG traces from female WT and SMNRKO mice pretreated with CYP or CYP and MK801. (M, N, O, P and Q) Summarized data of CMGs from female WT mice (CYP, n = 4; CYP-MK801, n = 4) and SMNRKO mice (CYP, n = 4; CYP-MK801, n = 4) indicate similar altered urodynamic function between WT mice and SMNRKO mice pretreated with CYP or CYP and MK801. MK801 reversed the decreased voiding interval, peak pressure, compliance, and increased basal pressure induced by CYP in both mice. Data are shown as box and whiskers, the centerline is the median of the data set, the box represents 75% of the data, and bars indicate whiskers from minimum to maximum. Data were analyzed by the Student t test. ∗P < 0.05, ∗∗P < 0.001 compared with normal female WT mice and control WT mice. n.s, No significance (>0.05).

Figure 6 Deficiency of NMDAR in BSM had no effects on pathological bladder BSM contractility.

(A, B, C and D) Representative traces of BSM contraction from female WT (CYP, n = 9; CYP-MK801, n = 9) and SMNRKO (CYP, n = 9; CYP-MK801, n = 9) mice pretreated with CYP or CYP and MK801in response to EFS (1, 2, 5, 10, 20, and 50 Hz). (E, F, G and H) Summarized data of EFS-induced contraction, KCl-induced contraction, carbachol-induced contraction, and a,b-meATP-induced contraction from SMNRKO mice pretreated with CYP or CYP and MK801 showed similar contraction force with WT mice pretreated with CYP or CYP and MK801. MK801 reversed reduced contraction force induced by CYP in both WT and SMNRKO mice. Data are shown as box and whiskers, the centerline is the median of the data set, the box represents 75% of the data, and bars indicate whiskers from minimum to maximum. Data were analyzed using the Student t test. ∗P < 0.05, ∗∗P < 0.001 compared with normal female WT mice and the control WT mice. n.s: No significance (>0.05).

Pharmacotherapies targeted on NMDAR in BSM did not improve pathological bladder function

Intravenous injection of MK801 is an effective way to suppress CYP-induced bladder overactivity (Liu et al., 2015). Whether the direct instillation of NMDAR antagonists into the bladder had the same effect was not known. Hence, CMG studies were performed during NMDAR antagonist infusion into CYP-induced OAB. The CYP ruptured and caused significant damage to the urothelium, thus allowing the infusion drug to penetrate into the smooth muscle layer and work on BSM cells. As shown in Fig. 7, up to 100 µM D-AP5, CGS 19755, and MK 801 instillation did not improve bladder dysfunction. The mice after receiving NMDAR antagonist instillation displayed similar urodynamic abnormalities compared with the mice before receiving instillation. Besides, the direct instillation of NMDAR agonists into the bladder failed to worsen bladder function in CYP-induced OAB (Fig. 7). These results were consistent with a recent report that revealed the NMDAR agonists, including NMDA and RS-tetrazol-5-yl glycine, at concentrations up to 100 µM had no effect on BSM contractile force, and NMDAR antagonists, including D-AP5, CGS19755, and MK801, had little or no effect on BSM contraction (Chen et al., 2020). Furthermore, the lack of NMDAR in BSM had no effects on bladder morphology and function under normal or pathological conditions, and SMNRKO mice with cystitis after MK801 treatment displayed no differences in bladder phenotypes compared with WT mice with cystitis after MK801 treatment (Figs. 4, 5 and 6). Taken together, these data suggested that pharmacological drugs targeted on NMDAR in BSM did not improve pathological bladder function.

Figure 7 Direct instillation of NMDAR agonists and antagonists into the CYP-induced cystitis bladder did not impact urodynamic function.

(A, B, C, D, E and F) Representative CMG traces from female WT mice pretreated with CYP during the bladder infused with NMDAR agonists and antagonists, including 100 µM NMDA (n = 4), 100 µM RS-tetrazol-5-yl glycine (n = 4), 100 µM D-AP5 (n = 4), 100 µM CGS 19755 (n = 4), and 100 µM MK 801 (n = 6). (G, H, I, J and K) Summarized data of CMGs indicate that NMDAR agonist or antagonist instillation did not impact urodynamic function. Data are shown as box and whiskers, the centerline is the median of the data set, the box represents 75% of the data, and bars indicate whiskers from minimum to maximum. Data were analyzed using Student t test. ∗P < 0.05, ∗∗P < 0.001 compared with control.

Discussion

The role of NMDAR is mainly known through its neural activity. However, its expression and role in a variety of nonneural peripheral cells and organs have also been shown. In particular, NDMAR is expressed throughout the cardiovascular system and plays a role in electrical and pacemaker activities (Hogan-Cann & Anderson, 2016). In the lung, ASM cells expressing NMDAR enhance airway responsiveness (Cukier, Cortina-Borja & Brading, 1997; Dumas et al., 2018; Quatredeniers et al., 2019) and contribute directly to bronchiole smooth muscle contraction (Antošová & Strapková, 2013; Anaparti et al., 2015). NMDAR subunit mRNA and proteins have also been found to be expressed in the lower urogenital tract and may participate in the control of organ tone for male sexual activity (Gonzalez-Cadavid et al., 2000). Emerging evidence shows that NMDAR is involved in the development of OAB, pharmacological inhibition of which markedly reduced bladder overactivity and improved bladder function (Liu et al., 2015). BSM from OAB often shows enhanced spontaneous contractile activity, which has been documented in human bladder strips from obstructed unstable bladders and patients with neuropathy (Sibley, 1984; German et al., 1995; Steers, 2002). NMDAR previously studied in smooth muscle is associated with airway remolding and contributes to airway hyperreactivity (Anaparti et al., 2015). This study investigated the roles and effects of NMDAR in BSM on the regulation of bladder morphology and function under normal physiological and pathological conditions. The inhibition of NMDAR by injection of MK801 had positive effects on bladder remodeling and bladder function, but the deficiency of NMDAR in BSM had no such effects. Furthermore, the study verified that NDMAR in BSM was not the pharmacotherapy target for CYP-induced cystitis.

Smooth muscle–specific Grin1 knockout mice were generated in the present study. Grin1 gene encodes GluN1, which is a subunit of NMDAR. Previous studies found that several NDMAR isoforms were expressed in human and mouse bladder (Yue et al., 2014; Fagerberg et al., 2014). The findings were consistent with a most recent study showing that BSM cells expressed NMDAR subunits, which were verified by the single-cell sequencing method (Yu et al., 2019). NMDAR is composed of four subunits: two obligatory subunits GluN1 and two variable subunits including GluN-2A/2B/2C/2D/3A/3B. Therefore, the knockout of obligatory subunit GluN1 in BSM theoretically created an NMDAR-null in BSM. This strategy was previously used in the lung to study the role of NMDAR in ASM (Dumas et al., 2018). The bladders from SMNRKO mice showed normal size and weight, an intact bladder wall with different layers, and well-arranged BSM bundles with normal size smooth muscle cells, indicating that the deficiency of NMDAR in BSM under normal physiological conditions did not impact BSM cells and change bladder morphology. Unexpectedly, SMNRKO mice had normal voiding patterns and urodynamic function, indicating normal bladder function. BSM contractility is essential for physiological bladder function (Hao et al., 2019; Chen et al., 2020). The study further found that BSM strips from SMNRKO mice exhibited normal contractility, which was consistent with previous findings that the NMDAR agonists and antagonists had little or no effect on BSM contraction (Chen et al., 2020), indicating that NMDAR in BSM did not regulate BSM contractility, which explained why the loss of NMDAR in BSM had normal bladder function. Consistently, Dumas also reported that no differences were found in vascular remodeling and pulmonary function under normoxic conditions between SMNRKO mice and WT mice (Dumas et al. 2019). NMDAR subunits might not be activated or form fully active receptors in BSM under normal physiological conditions, deficiency of which had no effects on bladder morphology and function.

NMDAR activation is commonly found to be associated with pathological conditions. For example, NMDAR activation was found to mediate visceral hyperactivity in irritable bowel syndrome and OAB (Gaudreau & Plourde, 2004; Yokoyama, 2010). In the cerebral or aortic endothelium, NMDAR activation contributed to endothelial barrier disruption, inflammatory cell infiltration, oxidative stress, and proliferation (Sharp et al., 2005; Chen et al., 2005; András et al., 2007; Reijerkerk et al., 2010). In aortic and ASM cells, NMDAR activation stimulated cell proliferation, promoted cardiac and vascular remodeling, and induced PAH(Qureshi et al., 2005; Doronzo et al., 2010; Anaparti et al., 2015; Dumas et al., 2018). In contrast, the targeted knockout of NDMAR in ASM had beneficial effects on cardiac and vascular remolding and improved PAH (Dumas et al. 2019). The aforementioned studies suggested that NDMAR activation played roles in pathological conditions, deficiency of which had beneficial effects. However, in the present study, both SMNRKO mice and WT mice showed marked histological changes within the bladder after CYP treatment. Unexpectedly, NMDAR deficiency in BSM did not improve pathological bladder remolding. Furthermore, CYP-treated SMNRKO mice displayed similar altered voiding patterns and urodynamic abnormalities and impaired BSM contractility as CYP-treated WT mice, suggesting that NMDAR deficiency in BSM did not improve pathological bladder function either. Liu et al. (2015) previously demonstrated that NDMAR activation was primarily present in the central nervous system but not in the bladder during CYP-induced cystitis, indicating that CYP-induced cystitis did not stimulate NMDAR activation in the bladder. Thus, NMDAR in BSM was not involved in CYP-induced pathological bladder morphology and function.

The pharmacological inhibition of NMDAR by the intravenous injection of MK801 has been previously reported to inhibit bladder overactivity caused by cerebral infarction (Yotsuyanagi et al., 2005) and attenuate hyperreflexia in the micturition reflex induced by crystalluria, partial bladder outlet obstuction (BOO), and nerve injury (Ishida et al., 2003; Tanaka et al., 2003; Yokoyama et al., 2004; Kontani & Ueda, 2005). In addition, a most recent study reported that the injection of MK801 reduced bladder hypertrophy and suppressed bladder overactivity induced by CYP (Liu et al., 2015), suggesting that MK801 was effective for OAB treatment in mice. Consistently, in the present study, the injection of MK801 reversed pathological bladder histology and rescued bladder dysfunction in CYP-treated WT mice and SMNRKO mice. However, WT mice still displayed a similar bladder morphology, voiding pattern, urodynamics, and BSM contractility as SMNRKO mice after MK801 treatment, suggesting that NMDAR in BSM was not the treatment target of MK 801 for CYP-induced cystitis, which was further supported by the results of CMGs during the direct instillation of NMDAR agonists or antagonists into the CYP-induced cystitis bladder. The direct instillation of NDMAR antagonists, including MK801, D-AP5, and CGS 19755, into the bladder did not correct pathological bladder urodynamics. The instillation of NMDAR agonists, including NMDA and RS-tetrazol-5-yl glycine did not worse the bladder urodynamics. These results were consistent with the findings of previous myography studies revealing that NMDAR agonists NDMA and RS-tetrazol-5-yl glycine had no effects on BSM contractile force, and NMDAR antagonists MK801, D-AP5, and CGS 19755 had little or no effects on BSM contraction (Chen et al., 2020). Besides, the deficiency of NMDAR in BSM did not contribute to bladder morphology and function under normal physiological and pathological conditions. Therefore, it was believed that the intravenous injection of MK801 improving pathological bladder morphology and function was not mediated by the inhibition of NMDAR in BSM. Yoshiyama, Roppolo & de Groat (1993) suggested that the inhibition of micturition reflex by MK801 was mediated by an action on the lumbosacral spinal cord. Liu et al. (2015) reported that the injection of D-AP5, which could not cross the blood–brain barrier and block central NMDAR activity, failed to reduce the pathological bladder morphology and function during CYP-induced cystitis. However, the injection of MK801, which crossed the blood–brain barrier and inhibited central NMDAR activity, could reduce bladder hypertrophy and improve bladder function in CYP-induced cystitis. Furthermore, cystitis-induced NMDAR activity was found to be extremely low in the bladder compared with the spinal cord (Liu et al., 2015). Thus, NMDAR in the central nervous system was a pharmacotherapy target of MK801 for CYP-induced cystitis treatment, but not NMDAR in BSM.

The present study showed that NMDAR in BSM was not essential for normal physiological and pathological bladder morphology or function, and MK801-mediated improvement in pathological bladder function was not associated with NMDAR in BSM.

Supplemental Information

Supplemental Information 1 Raw data

Click here for additional data file.

Supplemental Information 2 The ARRIVE guidelines 2.0: author checklist

Click here for additional data file.

Additional Information and Declarations

Competing Interests

Author Contributions

Animal Ethics

Data Availability

The authors declare there are no competing interests.

Xiang Xie conceived and designed the experiments, performed the experiments, analyzed the data, prepared figures and/or tables, authored or reviewed drafts of the paper, formal analysis; Methodology, and approved the final draft.

Chuang Luo, JiaYu Liang, Run Huang, Jia Li Yang, Linlong Li, YangYang Li and Hongming Xing performed the experiments, prepared figures and/or tables, and approved the final draft.

Huan Chen conceived and designed the experiments, performed the experiments, analyzed the data, prepared figures and/or tables, authored or reviewed drafts of the paper, funding acquisition; Supervision; Project administration, and approved the final draft.

The following information was supplied relating to ethical approvals (i.e., approving body and any reference numbers):

All animal care and experimental procedures were performed in adherence to the National Institutes of Health Guidelines for the Care and Use Committee of the Southwest Medical University (20180307002).

The following information was supplied regarding data availability:

The raw measurements are available in the Supplementary File.

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
