# Peer review of "NMDAR in bladder smooth muscle is not a pharmacotherapy target for overactive bladder in mice"

_PeerJ, doi:10.7717/peerj.11684_

## Round 0.1 · original submission · Major Revisions

Dear authors

Please change the manuscript according to the reviewers' suggestions.

Thanks

Best regards

Ferdinand Frauscher

Reviewer 1 ·

Basic reporting

no comment

Experimental design

no comment

Validity of the findings

no comment

Additional comments

In this study, Xie and co-authors used a smooth muscle-specific NMDAR knockout mouse to investigate the potential role of NMDAR in OAB, they found that NMDAR is not a pharmacotherapy target in BSM. The overall experimental design is excellent, results from this investigation are robust and well presented. The speculations of the results are reasonable and the conclusions based on theses results are convincing.
I only have one suggestion, which is related to the contractility results, mostly presented in Fig 3. Are the thickness of the muscle well controlled ? It can be very hard to match the width and thickness of the muscle strip during dissection. Based on the method section and it seem, in some experiments, the width of the muscle strip is not very well controlled. In that case, the authors should consider using stress instead of force to present contractility in Fig 3.

·

Basic reporting

no comments

Experimental design

no comments

Validity of the findings

no comments

Additional comments

General comments;
This study was comprehensively performed to confirm that there is no role of NMDAR in bladder smooth muscle in organization of normal bladder morphology and function as well as in the development of CPY-induced abnormal bladder morphology and bladder dysfunction. Although this study is a completely negative study, the experimental methods are reasonable and the results are clearly stated. This study would be a valuable literature in the field of investigation on the control of the lower urinary tract.


Specific comments;
Wordings should be revised at several sentences.
Line 52: nycturia → nocturia
Line 105: euthanasia was performed → Euthanasia was performed
Line 137: isolated strips → Isolated strips
Line 144: function. CMG → function, CMG
Line 340: a recent report that revealing → that revealed

Reviewer 3 ·

Basic reporting

In this manuscript, the authors use clear and professional English throughout. All the sections are well explained, and the results seem to address the claims made by the authors. Literature references are provided and seem to be in order. Article structure is acceptable, and figures are understandable.

Experimental design

The experimental design seems valid. Research question is well defined and relevant to the audience. There are technical discrepancies in the model which are discussed in greater detail in the relevant section of the review. Methods description is adequate.

Validity of the findings

Some of the major concerns in the manuscript are summarized below-

1. The authors discuss generation of SMNRKO mice. However, Is this F1 generation of mice that are used for this study? In this case the mice are still heterozygous. A tissue specific KO, which is needed for this manuscript as the authors plan to knockout SMRKO only in the bladder smooth muscle (BSM). For this, two rounds of breeding is generally necessary. The authors must state if this was so in the methods section.
2. The efficacy of SMNRKO is not validated by the authors. This is most important, especially because the authors find no difference in WT vs KO. It could be simply because there was no KO. The authors must present stainings of NMDAR in bladder smooth muscles to prove complete KO in BSM and as a control perform staining on other tissues from the same mice to show presence of NMDAR in other tissues. This will validate the KO model.
3. Authors must also present genotyping data of the KO mice vs wild type mice to analyze the KO site. Again, BSM as well as other control tissues must be used to isolate DNA.

Other minor concerns are as follows-
1. Statistical softwares that were used must be mentioned in the methods section.
2. In the experimental design section, full names of methods must be written instead of abbreviations. This will enhance readers' experience.
3. Many figures have missing statistical presentation on graphs. This has to be rectified.
4. It is generally a good idea to include figure legends in all the panels of figures. The authors have skipped this which may confuse the readers.
5. Box and whisker plots in Figure 5 have two sets of statistics. The authors must explain why?
6. In the same figure some plots have p values written on them while others have asterisks. Consistency must be maintained. If not significant ns should be written.
7. In the abstract line 37- it should be induced 'by' CYP.

---

## Round 0.2 · accepted · Accept

Very well performed review.

Thanks

Best regards

Ferdinand

Reviewer 1 ·

Basic reporting

no comment

Experimental design

no comment

Validity of the findings

no comment

Additional comments

The new data from authors suggests that the muscle thickness is indeed well control. My concerns have been addressed.

Reviewer 3 ·

Basic reporting

no comment

Experimental design

no comment

Validity of the findings

no comment

Additional comments

The authors have addressed all the major concerns in their re-submission. I would recommend the authors to present their genotyping data, mRNA expression and western blot (comment 2 response) as a supplementary figure for the benefit of readers.
I congratulate the authors on their excellent work.